# Estimating Winter Canola Aboveground Biomass from Hyperspectral Images Using Narrowband Spectra-Texture Features and Machine Learning

**DOI:** 10.3390/plants13212978

**Published:** 2024-10-25

**Authors:** Xia Liu, Ruiqi Du, Youzhen Xiang, Junying Chen, Fucang Zhang, Hongzhao Shi, Zijun Tang, Xin Wang

**Affiliations:** 1College of Water Conservancy and Civil Engineering, Inner Mongolia Agricultural University, Hohhot 010010, China; ruiqidu@nwafu.edu.cn; 2College of Water Resources and Architecture Engineering, Northwest A&F University, Yangling 712100, China; youzhenxiang@nwafu.edu.cn (Y.X.); junyingchen@nwafu.edu.cn (J.C.); zhangfc@nwsuaf.edu.cn (F.Z.); shihongzhao7@nwafu.edu.cn (H.S.); tangzijun@nwafu.edu.cn (Z.T.); wx123@nwafu.edu.cn (X.W.)

**Keywords:** UAV, hyperspectral, texture, biomass, narrowband

## Abstract

Aboveground biomass (AGB) is a critical indicator for monitoring the crop growth status and predicting yields. UAV remote sensing technology offers an efficient and non-destructive method for collecting crop information in small-scale agricultural fields. High-resolution hyperspectral images provide abundant spectral-textural information, but whether they can enhance the accuracy of crop biomass estimations remains subject to further investigation. This study evaluates the predictability of winter canola AGB by integrating the narrowband spectra and texture features from UAV hyperspectral images. Specifically, narrowband spectra and vegetation indices were extracted from the hyperspectral images. The Gray Level Co-occurrence Matrix (GLCM) method was employed to compute texture indices. Correlation analysis and autocorrelation analysis were utilized to determine the final spectral feature scheme, texture feature scheme, and spectral-texture feature scheme. Subsequently, machine learning algorithms were applied to develop estimation models for winter canola biomass. The results indicate: (1) For spectra features, narrow-bands at 450~510 nm, 680~738 nm, 910~940 nm wavelength, as well as vegetation indices containing red-edge narrow-bands, showed outstanding performance with correlation coefficients ranging from 0.49 to 0.65; For texture features, narrow-band texture parameters CON, DIS, ENT, ASM, and vegetation index texture parameter COR demonstrated significant performance, with correlation coefficients between 0.65 and 0.72; (2) The Adaboost model using the spectra-texture feature scheme exhibited the best performance in estimating winter canola biomass (R^2^ = 0.91; RMSE = 1710.79 kg/ha; NRMSE = 19.88%); (3) The combined use of narrowband spectra and texture feature significantly improved the estimation accuracy of winter canola biomass. Compared to the spectra feature scheme, the model’s R^2^ increased by 11.2%, RMSE decreased by 29%, and NRMSE reduced by 17%. These findings provide a reference for studies on UAV hyperspectral remote sensing monitoring of crop growth status.

## 1. Introduction

Aboveground biomass (AGB) refers to the total mass of vegetation above the ground per unit area and is closely related to crop yield [1]. Timely and accurate monitoring of crop AGB helps improve farmland management efficiency [2]. Traditional manual in-situ AGB measurement is time-consuming, laborious, destructive, and unable to provide continuous spatial distribution [3]. Compared to traditional measurement, optical remote sensing technology can obtain information on the crop physiological and biochemical properties by capturing crop canopy spectra, thus becomes a popular and promising monitoring method [4].

Satellites, Unmanned Aerial Vehicle (UAV), and near-ground remote sensing platforms can be utilized for the rapid, non-destructive acquisition of crop growth indicators such as AGB [5]. Compared to other platforms, UAV has the advantages of convenient takeoff, flexibility, and high image resolution, becoming an important technical mean for accurately obtaining field-scale crop information [6]. By performing ratios, linear combinations, or nonlinear combinations of spectral reflectance at different wavelengths, vegetation indices can be calculated [7]. Vegetation indices have always been a simple and effective method for quantitatively estimating vegetation growth parameters [8]. Consequently, a multitude of vegetation indices closely related to AGB have been developed, such as the Normalized Difference Vegetation Index (NDVI, [9]), Normalized Difference Matter Index (NDMI, [10]), Ratio Vegetation Index (RVI, [11]), Modified Simple Ratio (MSR, [12]), Modified Chlorophyll Absorption Ratio Index (MCARI, [13]), and Transformed Chlorophyll Absorption Reflectance Index (TCARI, [14]). Combining these indices with modeling algorithms can further enhance estimation accuracy. The commonly used modeling algorithms are mainly divided into two categories: machine learning algorithms, such as Random Forest Regression (RFR) and Support Vector Machine Regression (SVR) [15]; and traditional regression algorithms, such as Gaussian Process Regression (GPR) [16].

However, the capability of vegetation index on the crop AGB estimation is limited due to the saturation effect caused by dense canopy [17]. This phenomenon can be alleviated by using narrowband vegetation indices from hyperspectral images [18]. Besides spectra information, ultra-high ground resolution UAV images (1 cm or higher) provides abundant image texture information for crop phenotyping studies [8]. Image texture is a method of quantifying the variation in pixel values between adjacent pixels, effectively reflecting the physical properties of an object’s surface. Common texture parameters are derived from the Gray Level Co-occurrence Matrix (GLCM). These statistics describe the variations in pixel grayscale values within a given area and the relationships between pairs of pixels [19]. Liu et al. combined multispectral vegetation indices with texture parameter to achieve reasonable winter canola nitrogen nutrition index (NNI) estimation [20]. Compared to multispectral images, hyperspectral images provide more spectra information at narrow wavelengths. The corresponding texture parameters can be used to describe a specific object more objectively and reliably [21]. Current researches have mainly utilized the spectral information from hyperspectral images, with fewer studies focus on the use of hyper-spectral images texture features on assessing crop traits.

Canola is one of the important oil crops in China. Under the background of global climate change, the planting range of canola is showing a trend of moving northward and widely planted in the northwest region. For this, we aim to investigate the potential of ultra-high spatial resolution UAV hyperspectral images in estimating winter canola biomass: (1) Evaluate the capability of narrowband spectral-texture features based on UAV hyperspectral images in estimating winter canola biomass; (2) Validate and compare the reliability of three feature scheme (narrowband spectra feature scheme, narrowband texture feature scheme, and narrowband spectral-texture feature scheme) in estimating winter canola biomass; (3) Determine the best biomass estimation model for narrowband spectral-texture features.

## 2. Results

### 2.1. Sensitive Analysis of Spectra-Texture Features to Canola AGB

By UAV hyperspectral images, spectral-texture information from 138 narrowband bands and 21 narrowband vegetation indices was obtained. Correlation analysis was used to evaluate the capability of these data in characterizing winter canola AGB (Figure 1). For the correlation coefficients, the capabilities was ranked as follows: narrowband vegetation index (*r* = 0.51~0.65) > texture index from narrowband vegetation index (*r* = 0.13~0.72) > texture index from narrowband bands (*r* = 0.10~0.68) > narrowband spectral reflectance (*r* = 0.18~0.64). Vegetation indices enhance the ability to characterize winter canola biomass by normalizing multiple narrowband spectral and texture information. For narrowband bands, the performance of narrowband at 450~510 nm, 680~738 nm, and 910~940 nm wavelength is relatively prominent, with correlation coefficient of 0.49~0.64; for narrowband vegetation indices, those using the red-edge narrowband (700~800 nm) perform relatively well, with correlation coefficients of 0.62~0.65; for texture index from narrowband bands, CON, DIS, ENT, and ASM show relatively good performance, with correlation coefficients of 0.65~0.68; for texture parameters from narrowband vegetation index, COR stands out, with correlation coefficients of 0.67~0.72.

### 2.2. Autocorrelation Analysis of Spectra-Texture Features

As shown in Figure 1, we have statistically analyzed a total of 159 spectral feature variables and 954 texture feature variables. When the number of feature variables is much larger than the number of samples, it becomes difficult for models to fully learn the highly nonlinear complex relationships between variables. To address this, variables were screened using a correlation coefficient threshold of 0.6, resulting in a final count of 18 spectral feature variables and 44 texture feature variables. For narrowband spectral features, the selected variable was Band_726_, Band_718_, Band_734_, Band_690_, Band_498_, Band_686_ and Band_502_; For narrowband spectral indices, the selected variables were DNCI, WDRVI, MSR, MSAVI, NDI, NDRE, MTCI, CI, TCARI, mrNDVI and VOGREI3; For texture index from narrowband, the selected variables were mainly concentrated in DIS and ASM (470~520 nm). For the texture index from narrowband vegetation index, the selected variables were mainly focused on the COR. The corresponding indices were MNDVI, SRVI2, ARVI, TCARI, NDVI, NDRE, DCNI, NDI, VOGREI3, and WDRVI. Compared to similar studies, the number of selected variables is still relatively large. To avoid potential multi-collinearity among variables that could adversely affect model stability, autocorrelation analysis was used to further screen the selected variables, with the results shown in Figure 2.

For spectral feature variables, DNCI generally has a weaker connection with other variables and exhibits strong independence. mrNDVI and NDI have stronger collinearity with other variables, with correlation coefficients ranging from 0.28 to 0.91. After comparison, the optimal combination of spectral feature variables was determined to be Band_498_, Band_502_, Band_734_, DNCI, MSR, MTCAI, CI and TCARI. For texture feature variables, the collinearity phenomenon is evident in DIS and ASM, with correlation coefficients ranging from 0.39 to 0.96. In contrast, the COR texture parameter from narrowband spectral indices shows stronger independence. After comparison, the optimal combination of texture feature variables was determined to be CON_686_, CON_698_, CON_678_, CON_670_, CON_662_, ENT_482_, ENT_494_, ENT_470_, ASM_534_, VOGREI3_CON_, OSAVI_DIS_, OSAVI_HOM_, MNDVI_COR_, SRVI2_COR_, ARVI_COR_, TCARI_COR_, NDVI_COR_, NDRE_COR_, DCNI_COR_, NDI_COR_, VOGREI3_COR_, WDRVI_COR_ and NDVI_ASM_. Accordingly, the number of variables in the spectra feature scheme, texture feature scheme, and spectral-texture feature scheme was 8, 23, and 31, respectively.

### 2.3. Evaluation of Estimation Accuracy in Canola AGB

Figure 3 illustrates the accuracy of estimating rapeseed biomass using different machine learning algorithms and feature variables. Overall, the accuracy ranking of the three schemes is as follows: spectra-texture feature scheme (R^2^ = 0.43~0.91; RMSE = 75.93~3645.81 kg/ha; NRMSE = 18.16~29.23%) > texture feature scheme (R^2^ = 0.41~0.89; RMSE = 73.37~4645.81 kg/ha; NRMSE = 18.40~31.81%) > spectra feature scheme (R^2^ = 0.38~0.84; RMSE = 77.54~4199.59 kg/ha; 17.21~30.31%); The accuracy ranking of the four growth stages is as follows: overwintering (R^2^ = 0.44~0.77; RMSE = 73.37~111.24 kg/ha; NRMSE = 19.27~29.42%) > bolting (R^2^ = 0.43~0.62; RMSE = 564.58~965.26 kg/ha; NRMSE = 17.21~29.23%) > flowering (R^2^ = 0.38~0.59; RMSE = 1633.19~2848.63 kg/ha; NRMSE = 18.16~30.31%) > podding (R^2^ = 0.47~0.71; RMSE = 2873.52~4645.82 kg/ha; NRMSE = 19.41~31.82%). As an image attribute dataset, texture features plays an indispensable role in estimating the winter canola AGB. Compared to spectra feature, texture feature has a clear advantage during the flowering and podding stages. This indicates that texture information has a unique advantage in interpreting dense canopy information. The combined use of spectral and texture information helps to decrease the error caused by spectra saturation.

The number of features may be an important factor affecting the performance of the algorithm. In the spectra feature scheme, the six algorithms are ranked as follows: Xgboost (R^2^ = 0.41~0.84; RMSE = 77.54~3816.26 kg/ha; NRMSE = 17.42~26.01%) > GBRT (R^2^ = 0.41~0.84; RMSE = 82.69~4040.94 kg/ha; NRMSE = 17.22~27.59%) > Adaboost (R^2^ = 0.46~0.82; RMSE = 94.82~3998.51 kg/ha; NRMSE = 22.60~27.28%) > RF (R^2^ = 0.38~0.84; RMSE = 86.93~3946.03 kg/ha; NRMSE = 22.79~26.92%) > SVR (R^2^ = 0.39~0.78; RMSE = 93.55~4012.57 kg/ha; NRMSE = 19.76~27.38%) > GPR (R^2^ = 0.39~0.83; RMSE = 112.44~4199.59 kg/ha; NRMSE = 23.61~30.32%); In the texture feature scheme, the ranking is as follows: Adaboost (R^2^= 0.51~0.89; RMSE = 73.37~3245.77 kg/ha; NRMSE = 18.40~22.02%) > RF (R^2^ = 0.51~0.85; RMSE = 86.45~3120.87 kg/ha; NRMSE = 19.79~22.97%) > GBRT (R^2^ = 0.46~0.84; RMSE = 88.05~3416.86 kg/ha; NRMSE = 23.08~25.58%) > Xgboost (R^2^ = 0.48~0.81; RMSE = 86.00~3619.82 kg/ha; NRMSE = 23.10~26.49%) > SVR (R^2^ = 0.54~0.71; RMSE = 91.59~3012.84 kg/ha; NRMSE = 20.38~29.06%) > GPR (R^2^ = 0.41~0.70; RMSE = 86.48~4645.82 kg/ha; NRMSE = 22.67~31.82%); In the spectral-texture feature scheme, the ranking is as follows: GBRT (R^2^ = 0.49~0.86; RMSE = 79.51~3005.50 kg/ha; NRMSE = 19.63~26.17%) > RF (R^2^ = 0.51~0.89; RMSE = 84.19~2924.98 kg/ha; NRMSE = 19.78~22.08%) > Xgboost (R^2^ = 0.48~0.85;RMSE = 88.41~3044.32 kg/ha; NRMSE = 19.28~23.65%) > Adaboost (R^2^ = 0.41~0.84; RMSE = 77.54~3816.26 kg/ha; NRMSE = 17.42~26.01%) > GPR (R^2^ = 0.43~0.77; RMSE = 92.14~3645.81 kg/ha; NRMSE = 23.41~27.02%) > SVR (R^2^ = 0.51~0.62; RMSE = 91.57~3239.99 kg/ha; NRMSE = 18.83~29.06%). As the number of input variables increases, the nonlinear relationships between variables become more complex. Compared with traditional machine learning algorithms (i.e., SVR and GPR), ensemble learning (i.e., RF, Xgboost, Adaboost, and GBRT) can improve estimation accuracy.

For each scheme, the optimal algorithm was used to estimate the winter canola AGB throughout the entire growth period, and the estimation results are shown in Figure 4. Overall, the measured and the estimated AGB was close to the 1:1 line. The model can accurately estimate winter canola AGB during the overwintering period. From the bolting stage, the model’s estimation deviation becomes increasingly apparent. The deviation is mainly concentrated in the high AGB values (9000~15,000 kg/ha). This range corresponds to the flowering and podding growing stages. The overly dense canopy leads to the model’s overestimation of winter canola AGB. Compared with other feature schemes, the combined use of spectral and texture features helps to narrow the gap between actual and estimated AGB, but this advantage has limitations in the later growth stages of winter canola.

## 3. Discussion

### 3.1. Advantage of Texture Features

In this study, experimental fields with different water-nitrogen treatments were used to represent the actual filed. This will result in different winter canola AGB values. Sharma et al. (2022) has studied the impact of vegetation canopy structure on spectral reflectance [22]. The benefit of optical vegetation indices to estimate crop above-ground biomass has been verified. The winter canola AGB gradually increases with the advancement of the growth period, but the sensitivity of the vegetation index gradually decreases. For example, from the bolting to the flowering growth stage, AGB increases from 2605~3100 kg/ha to 8831~10,241 kg/ha. However, the correlation coefficient between MNDVI and AGB decreases from 0.64 to 0.36. Previous studies have also shown that the performance of vegetation indices is limited under dense canopy conditions [23]. Compared with traditional vegetation indices, the use of near-infrared vegetation indices with non-linear combinations helps to improve this situation. For example, the correlation coefficient between DCNI and the winter wheat at the flowering stage is 0.49. However, it cannot be denied that crop biomass estimation using only visible and near-infrared bands provided by optical sensors still poses certain challenges.

The canopy image of winter canola consists of complex soil, leaves, stems, shaded leaves, flowers, and pods. Thus, the high-resolution image of the winter canola contains rich texture information. Texture indices calculated from the gray-level co-occurrence matrix can fully characterize this feature [19]. Compared with optical vegetation indices, image texture parameters are more sensitive to the response of the winter canola canopy structure. The performance of the COR texture parameter is usually better than other texture parameters (Figure 1). As the winter canola grows, the canopy cover continues to increase, and the canopy structure becomes more complex. By comparison, the advantage of texture parameters is significant during the flowering and podding stages (Figure 5). This finding is similar to the role of texture in estimating the biomass of kiwifruit at the later growth stages [19]. It should be emphasized that in terms of model estimation performance, the combination of image texture parameters and vegetation indices can obtain more accurate estimates of winter canola AGB (Figure 1). This is because, when estimating samples across multiple growth stages, using image texture parameters independently can provide a more stable estimate than vegetation indices, overcoming the problem of estimation bias caused by the saturation of optical vegetation indices [24]. The combined use of image texture parameters and vegetation indices plays a complementary role in characterizing the vegetation canopy structure. The effectiveness of this approach has been confirmed in previous studies on other traits of crops (such as leaf area index), providing a new perspective and method for estimating the winter canola AGB [25].

### 3.2. Advantage of Narrowband Spectra Information

Research on estimating crop growth parameters using optical images has been widely carried out. In terms of sensor types, studies have largely focused on assessing the application potential of multispectral satellite and unmanned aerial vehicle (UAV) images. Although the cost of acquiring hyperspectral images is high compared to multispectral images, they can provide richer spectra information. More importantly, the use of UAV hyperspectral images can further enhance the use of texture information.

In this study, we explored the application potential of the spectral-texture information provided by 138 bands of hyperspectral images (450~950 nm) in estimating the winter canola AGB. The results show that using narrowband spectral-texture information can achieve relatively ideal accuracy (Figure 4 and Figure 5). At the same time, we compared this result with the performance of multispectral broadband bands (Figure 6). Broadband bands can be obtained through a weighted average of narrowband bands, and the wavelength range of each band refers to the DJI Phantom 4 UAV [26]. Figure 6 shows that the ability of broadband to capture changes in winter canola AGB is lower than that of narrowband. For example, the correlation coefficients for the broadband and narrowband spectra texture of the blue band are concentrated in the ranges of 0.49–0.63 and 0.50~0.69, respectively (Figure 6a). Narrowband spectral information can obtain the most sensitive areas of changes in winter rapeseed biomass, while broadband spectral information will blur the relationship between the two. For example, for the Red-edge band, the correlation coefficient between broadband spectra and canola AGB is 0.55. Compared to broadband spectra, the correlation coefficient between narrowband spectra with a wavelength of 726 nm and Canola AGB is 0.65 (Figure 6d). This further increases the gap between hyperspectral narrowband vegetation indices and multispectral broadband vegetation indices. At the same time, narrowband bands can construct more vegetation indices, enhancing the response relationship between spectral subtle changes and winter canola biomass (*r* = 0.52~0.66, Figure 2).

### 3.3. Scale Effect from UAV High Resolution Images

Previous studies have reported that the spatial resolution of images has a non-negligible impact on remote sensing estimation targets [27]. This is because, in addition to the complex relationship and heterogeneity between spectra and the objects, uneven lighting (low cloud effects), solar illumination (shadows and partially sunny areas), and the canopy structure of dense vegetation can affect the overall performance of spectral information [28]. In fact, these factors are closely related to the pixel size. Previous studies have considered and analyzed the impact of pixel size on the estimation of surface species classification, vegetation cover, and other targets [29]. Figure 7 shows the correlation analysis results between spectral texture information and the biomass of winter rapeseed at different resolutions. As the pixel size increases, the proportion of sunlight-illuminated vegetation, shaded vegetation, soil, and shaded soil contained in the pixel spectral information changes [30]. This is the fundamental reason for the relative change in the relationship between spectral-texture information and the winter canola AGB. Overall, the performance of vegetation indices shows weakening trend as the spatial resolution decreases (Figure 7). Previous studies can also confirm this [27,28,29]. However, a lower resolution will reduce the flight time of the UAV. When UAVs are applied to larger farmlands, both flight time and spatial resolution need be considered simultaneously. For winter canola, 2–4 cm resolution images can ensure the availability of training samples at the cost of some estimation accuracy. It should be noted that in this study, the coarse-resolution remote sensing images were obtained by resampling the original images, and these spectral reflectance values may not be the same as those obtained by flying at the actual set height. At a given flight height, mixed pixels in coarse-resolution images may further amplify scale effects, leading to unsolvable artifacts and distortions [31]. This will result in real coarse-resolution images performing lower than the results shown in Figure 7. Therefore, the optimization threshold for reducing resolution and increasing flight height needs to be further clarified.

### 3.4. Summary and Future

In this study, we evaluated the potential application of narrowband spectra-texture information provided by hyperspectral images in estimating the winter canola aboveground biomass. Compared to the broadband spectra-texture information provided by widely used multispectral images, hyperspectral images have clear advantages in improving the accuracy of aboveground biomass estimation, providing an important data source for extracting temporal characteristics of aboveground biomass during winter canola growth, and can be further applied to winter canola yield prediction and field irrigation-nitrogen supply management evaluation. However, the estimation models established by machine learning driven spectra-texture information also have certain potential application limitations. On the one hand, data-driven methods have weak theoretical rationality, and the established estimation models lack universality and are only applicable to specific samples. This results in the selected optimal spectra-texture feature combination and modeling algorithm not necessarily being applicable to all crops. When the observation conditions (e.g., sun-sensor-target geometric optical conditions, canopy structure, and soil background) change, it is necessary to re-determine the appropriate features combination and modeling methods. On the other hand, the robustness of the estimation model requires a certain number of samples. For low altitude remote sensing, collecting sufficiently representative samples on small-scale farmland is more difficult than that at regional scales. In the future, it is necessary to further explore the applicability of transfer learning strategies in extracting crop aboveground biomass information from low altitude remote sensing to reduce the modeling samples demand. Specifically, it is possible to attempt pre training estimation models using existing publicly available large-scale farmland datasets or theoretical spectra datasets, and then transferring them to estimate farmland crop aboveground biomass.

## 4. Material and Methods

### 4.1. Experiment Site

Field experiment (October 2021 to June 2022) on winter canola was conducted at the water-saving irrigation station of Northwest A&F University (108°24′ E, 34°18′ N). The planting density of the plants was 120,000 plants/ha. Climate and soil parameters of the experimental site can be found in [32]. The experiment set up two factors: irrigation amount and nitrogen application rate, corresponding 15 water-nitrogen treatments (45 plots, with each treatment repeated three times). The specific quantities and methods of application can be found in [32].

### 4.2. Data Collection

#### 4.2.1. Canola Above Biomass

Five representative plants were selected from each plot for destructive sampling, which could represent the overall growth of winter canola in the entire plot. After removing the surface dirt from the plants, they were placed in an oven and blanched at 105 °C for 30 min, followed by drying the crops to a constant weight at a constant temperature of 70 °C. The dry matter weight of the winter canola plants was measured using an electronic scale, and then converted into aboveground biomass (kg/ha) based on the planting density.

#### 4.2.2. UAV Hyperspectral Image

During the main growth stages of winter canola (overwintering, bolting, flowering, and podding stages), UAV hyperspectral images of the experimental plots were acquired under clear and cloudless weather conditions. The used UAV was the M300RTK produced by DJI company, Shenzhen, China, and the used hyperspectral camera was the Cubert UHD185 (Cubert company, Ulm, Germany). Image pre-processing was accomplished using Cubert Utils Touch and ENVI 5.3 software.

### 4.3. Biomass Estimation Model

#### 4.3.1. Spectral Feature

From the preprocessed hyperspectral images, 138 narrowband spectral reflectance features (450~1000 nm) were extracted. At the same time, 21 common narrowband vegetation indices were selected, with their calculation formulas and reference sources shown in Table 1.

#### 4.3.2. Texture Feature

The Gray Level Co-occurrence Matrix (GLCM) is a statistical data-based method for extracting image texture features [48]. In this study, we used Python software to calculate six texture parameters (CON, DIS, HOM, ENT, COR, and ASM) corresponding to each of the 138 narrowband spectral reflectance features and 21 narrowband vegetation indices. Specifically, first, use the Python software OpenCV library to read the grayscale images of 138 bands; Secondly, use the numpy library to calculate 21 vegetation indices and generate corresponding grayscale images; Finally, use the numpy library and GLCM texture parameter calculation formula to obtain the texture parameters of each grayscale image. The calculation formulas for these texture parameters can be obtained from [48]. In total, 954 (159 × 6) texture parameters were generated.

#### 4.3.3. Modeling Method

To fully evaluate the capabilities of narrowband spectral-texture information, we established three feature datasets using: spectra feature scheme, texture feature scheme, and spectra-texture feature scheme. The feature scheme was determined by sensitive analysis (Pearson correlation coefficient, *r*) between spectra/texture features and canola AGB. For each scheme, six machine learning algorithms (Adaboost, Xgboost, RF, GBRT, SVR, and GPR) were used to establish estimation models using Python 3.8. These algorithms are implemented by the scikits.learn toolkit provided by Python software. A total of 180 samples were collected. K-fold cross validation is used to avoid overfitting problems encountered during model training. The ratio of the number of samples in the modeling set to the validation set was 7:3. The estimation accuracy of the models was evaluated using the coefficient of determination (R^2^), root mean square error (RMSE), and normalized mean squared relative error (NMRSE) [49]. The technical roadmap of the entire study was shown in Figure 8.

## 5. Conclusions

(1)The narrowband spectral-texture features provided by UAV hyperspectral images are closely related to the winter canola AGB, with correlation coefficients concentrated between 0.34 and 0.68. In spectra features, the 450–510 nm, 680–738 nm, and 910–940 nm wavelength spectra and vegetation indices containing the red-edge wavelength perform prominently, with correlation coefficients of 0.49 to 0.65; in texture features, narrowband texture parameters of CON, DIS, ENT, ASM, and vegetation index texture parameters of COR perform prominently, with correlation coefficients of 0.65 to 0.72;(2)After variable selection through autocorrelation analysis, 8 spectra feature variables and 23 texture feature variables were identified as the optimal spectral-texture feature combination, with correlation coefficients ranging from 0.63 to 0.72;(3)Machine learning methods can establish robust models for estimating the winter canola AGB. The accuracy ranking of the six algorithms is as follows: Adaboost (R^2^ = 0.82~0.91; RMSE = 1710.79~2408.43 kg/ha; NRMSE = 19.88~24.07%) > RF (R^2^ = 0.84~0.89; RMSE = 1993.29~2392.10 kg/ha; NRMSE = 21.58~23.97%) > GBRT (R^2^ = 0.84~0.87; RMSE = 2274.08~2296.47 kg/ha; NRMSE = 23.26~23.40%) > Xgboost (R^2^ = 0.81~0.85; RMSE = 2246.95~2328.48 kg/ha; NRMSE = 23.10~23.64%) > GPR (R^2^ = 0.64~0.83; RMSE = 2351.82~2900.25 kg/ha; NRMSE = 23.73~27.02%) > SVM (R^2^ = 0.62~0.78; RMSE = 2622.88~3239.99 kg/ha; NRMSE = 25.35~29.06%);(4)The Adaboost model using the spectral-texture feature scheme performed the best in estimating the canola AGB. The combined use of narrowband spectra and image texture significantly improves the estimation accuracy. Compared to the spectra feature scheme, the model’s R^2^ increased by 11.2%, RMSE decreased by 29%, and NRMSE decreased by 17%.

## Figures and Tables

**Figure 1 plants-13-02978-f001:**
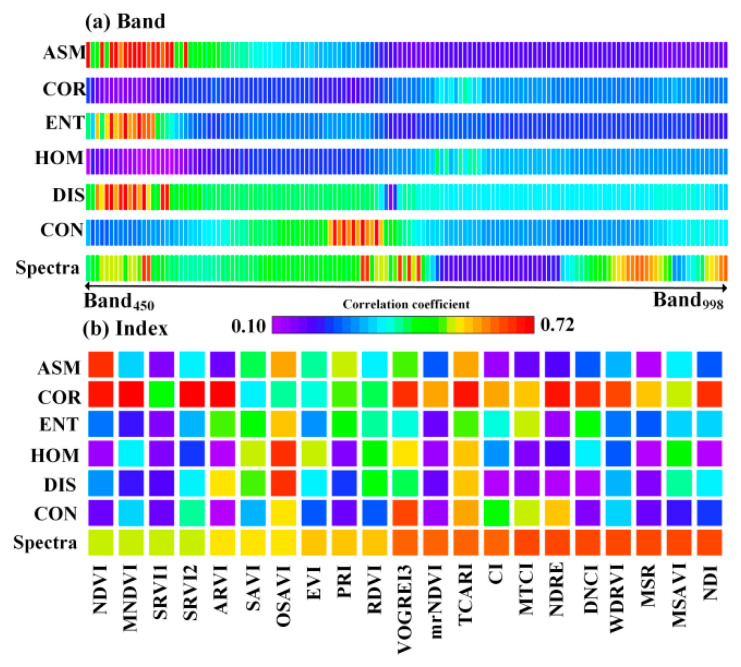
Correlation analysis between spectral-texture information and winter canola AGB. (**a**) Correlation analysis between 138 bands spectral reflectance/texture indicators and winter canola AGB; (**b**) Correlation analysis between 21 narrowband spectral indices/texture indicators and winter canola AGB.

**Figure 2 plants-13-02978-f002:**
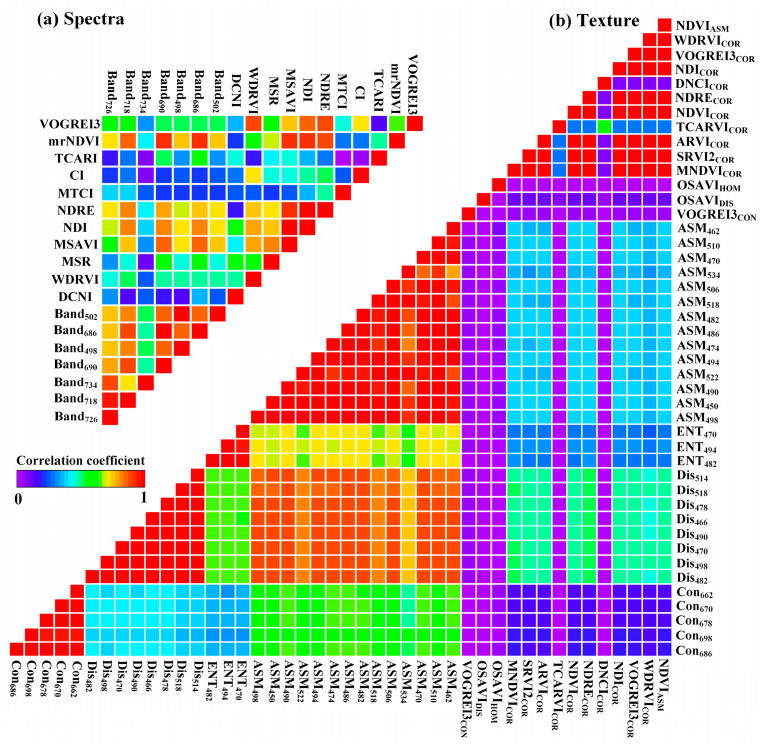
Autocorrelation analysis of (**a**) spectral and (**b**) texture features.

**Figure 3 plants-13-02978-f003:**
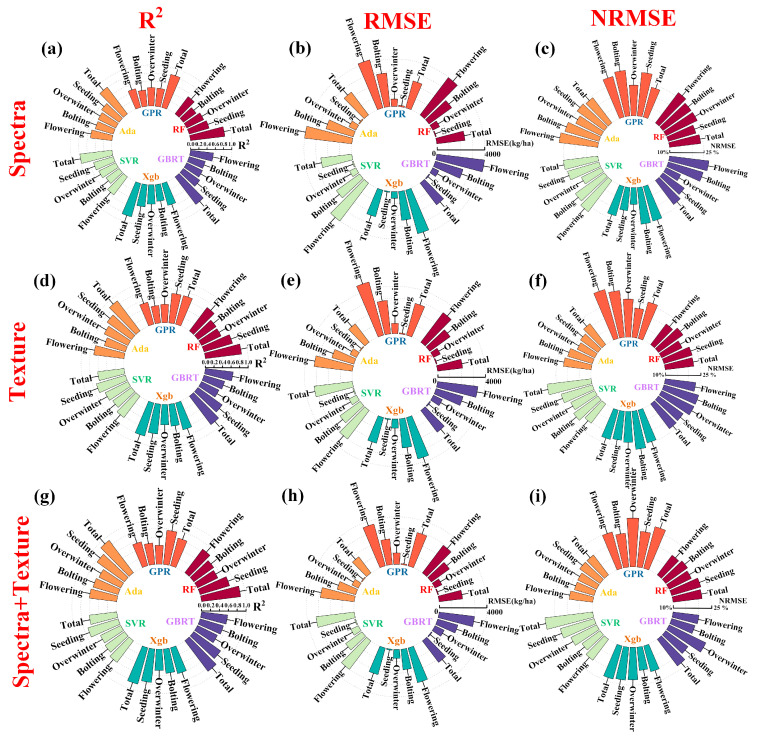
Evaluation of the accuracy of winter canola AGB estimation. (**a**–**c**), (**d**–**f**), (**g**–**i**) represent the evaluation of spectra, texture, and spectra-texture information in winter canola AGB estimation; (**a**)/(**d**)/(**g**), (**b**)/(**e**)/(**h**), (**c**)/(**f**)/(**i**) represent the estimation accuracy evaluation indicators R^2^, RMSE, and NRMSE, respectively. Six colors are used to represent modeling algorithm types.

**Figure 4 plants-13-02978-f004:**
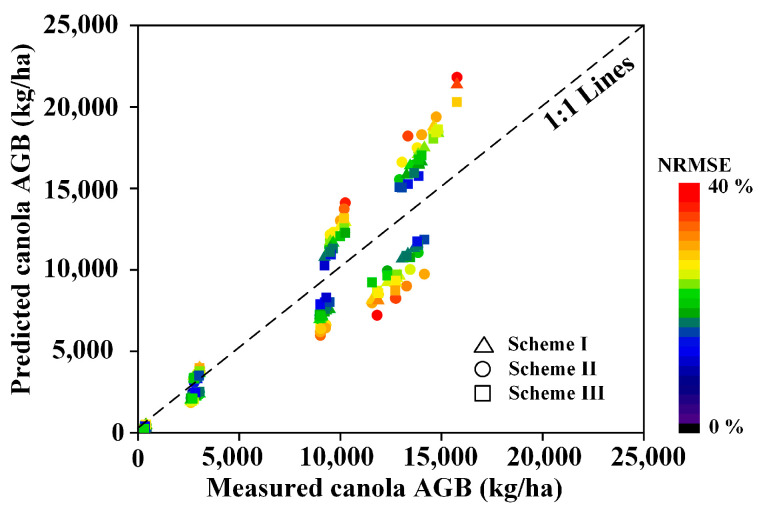
Scatter plot of estimated and measured winter canola AGB values.

**Figure 5 plants-13-02978-f005:**
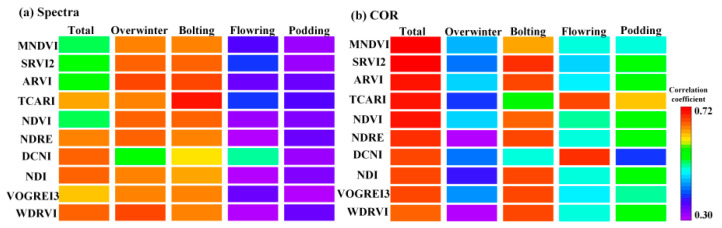
Comparison of the correlation between (**a**) spectra/(**b**) texture and winter canola AGB.

**Figure 6 plants-13-02978-f006:**
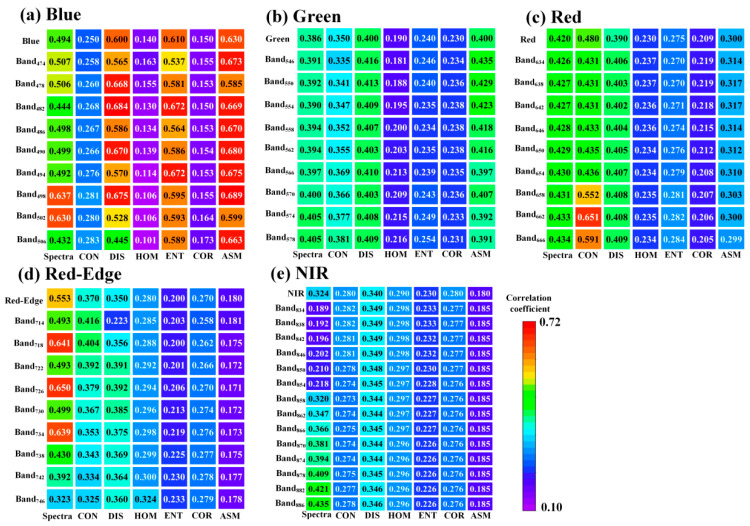
Comparison of the correlation between narrowband, broadband, and biomass.

**Figure 7 plants-13-02978-f007:**
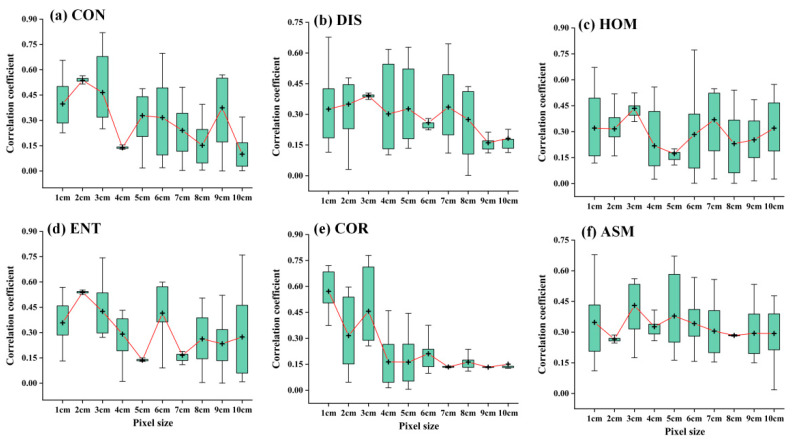
Effect of the pixel size on winter canola AGB estimation. + represents the average value of the correlation coefficient. The red connecting line is used to enhance the display of the variation of the correlation coefficient at different resolutions.

**Figure 8 plants-13-02978-f008:**
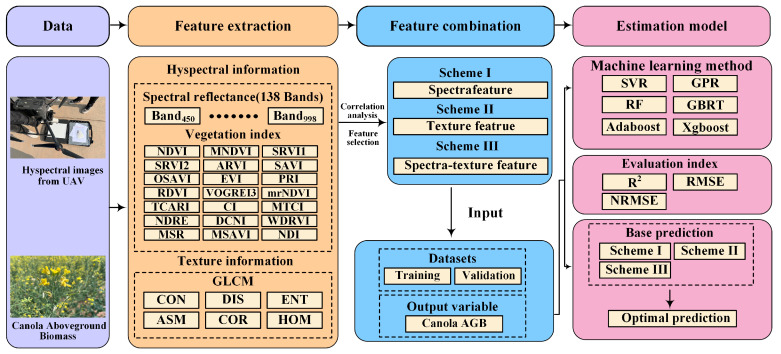
Technology roadmap.

**Table 1 plants-13-02978-t001:** The vegetation index.

Vegetation Index	Formula	References
Normalized difference vegetation index (NDVI)	(*R*_797_ − *R*_669_)/(*R*_797_ + *R*_669_)	[9]
Modified Normalized difference vegetation index (MNDVI)	(*R*_754_ − *R*_740_)/(*R*_754_ + *R*_740_)	[33]
Simple ratio vegetation index (SRVI1)	*R*_797_/*R*_669_	[11]
Simple ratio vegetation index (SRVI2)	*R*_754_/*R*_712_	[33]
Atmospherically resistant vegetation index (ARVI)	(*R*_797_ − 2*R*_669_ + *R*_454_)/(*R*_797_ + 2*R*_669_ − *R*_454_)	[34]
Soil-adjusted vegetation index (SAVI)	(1 + 0.5)(*R*_797_ − *R*_669_)/(*R*_797_ + *R*_669_ + 0.5)	[35]
Optimization of soil-adjusted vegetation index (OSAVI)	(1 + 0.16)(*R*_797_ − *R*_669_)/(*R*_797_ + *R*_669_ + 0.16)	[36]
Enhanced vegetation index (EVI)	2.5(*R*_797_ − *R*_669_)/(*R*_797_ + 6*R*_669_ − 7.5*R*_454_ + 1)	[35]
Photochemical Reflectance Index (PRI)	(*R*_531_ − *R*_570_)/(*R*_531_ + *R*_570_)	[37]
Renormalised Difference Vegetation Index (RDVI)	(*R*_800_ − *R*_670_)/(*R*_800_ + *R*_670_)^1/2^	[38]
Vogelman Red Edge Index 3 (VOGREI3)	(*R*_734_ − *R*_747_)/(*R*_715_ + *R*_720_)	[39]
Modified Red Normalized Difference Vegetation Index (mrNDVI)	(*R*_750_ − *R*_705_)/(*R*_750_ + *R*_705_ − 2*R*_445_)	[40]
Transformed Chlorophyll Absorption Reflectance Index (TCARI)	3((*R*_700_ − *R*_670_) − 0.2(*R*_700_ − *R*_550_)(*R*_700_/*R*_670_))	[14]
Chlorophyll Index (CI)	*R*_880_/*R*_590_ − 1	[41]
MERIS terrestrial chlorophyll index (MTCI)	(*R*_760_ − *R*_720_)/(*R*_720_ − *R*_670_)	[42]
Normalized difference red-edge (NDRE)	(*R*_790_ − *R*_720_)/(*R*_790_ + *R*_720_)	[43]
Double-peak canopy nitrogen index (DCNI)	(*R*_760_ − *R*_720_)/(*R*_720_ − *R*_670_)/(*R*_720_ − *R*_670_ + 0.09)	[44]
Wide dynamic range vegetation index (WDRVI)	(0.1*R*_800_ − *R*_670_)/(0.1*R*_800_ + *R*_670_)	[45]
Modified Simple Ratio (MSR)	(*R*_800_ − *R*_670_ − 1)/((*R*_800_ + *R*_670_)0.5 + 1)	[12]
Modified Soil Adjusted Vegetation Index (MSAVI)	*R*_800_ + 0.5 − ((*R*_800_ + 0.5)^2^ − 2(*R*_800_ − *R*_670_))^0.5^	[46]
Normalized Difference Index (NDI)	(*R*_850_ − *R*_710_)/(*R*_850_ + *R*_680_)	[47]

## Data Availability

Data are contained within the article.

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
