# Peer review of "Estimating Winter Canola Aboveground Biomass from Hyperspectral Images Using Narrowband Spectra-Texture Features and Machine Learning"

_plants, 2024, doi:10.3390/plants13212978_

Round 1

Reviewer 1 Report

Comments and Suggestions for Authors

1) This abstract effectively communicates the study's goal, methodology, and key findings in a concise manner. However, some areas could benefit from refinement for clarity, coherence, and flow. Some technical acronyms (e.g., GLCM, ASM) might benefit from brief explanations or omission to avoid confusion. 

2) Some portions of the results section is somewhat repetitive, especially when describing the performance of models. For instance, the reference to narrowband spectral features and their correlation coefficients might be streamlined.

3) Ln 102, I am concerned that the five plants should be randomly selected. How the representativeness was determined? 

4) Ln 124, In addition to the cited reference, describe the method of extracting image texture features here. 

5) The discrepancy observed at high AGB values may be influenced by the sampling technique employed in this study, as only five plants were selected from each plot. I suggest including a more detailed explanation of this deviation in the discussion section to better clarify the model's performance in predicting higher AGB values. Additionally, Fig. 5 shows no clear trend of underestimation or overestimation, which implies that random sources of error may play a more significant role, please clarify. 

Author Response

We thank the editors and reviewers for their hard work on this manuscript (plants-3166826). Your suggestions help to improved the quality of the manuscript. The parts of the manuscript that were revised have been marked in red. For the modified parts, the authors provide the following detailed explanation:

  • This abstract effectively communicates the study's goal, methodology, and key findings in a concise manner. However, some areas could benefit from refinement for clarity, coherence, and flow. Some technical acronyms (e.g., GLCM, ASM) might benefit from brief explanations or omission to avoid confusion.

Answer:

Thanks your suggestion !

The full names of professional terms have been added to the abstract, and the abbreviations are included in parentheses

Specifically, as follows:

“...The Gray Level Co-occurrence Matrix (GLCM) method was employed to compute six texture parameters: Contrast、Dissimilarity、Homogeneity、Entropy、Correlation、Angular Second Moment(CON, DIS, HOM, ENT, COR, and ASM)...”

“... Subsequently, six machine learning algorithms, Adaptive Boosting、Extreme Gradient Boosting、Random Forest、Gradient Boosting Regression Tree、Support Vector Regression and Gaussian Process Regression (Adaboost, Xgboost, RF, GBRT, SVR, and GPR), were applied to develop estimation models for winter canola biomass...”

2) Some portions of the results section is somewhat repetitive, especially when describing the performance of models. For instance, the reference to narrowband spectral features and their correlation coefficients might be streamlined.

Answer:

Thanks your suggestion !

These coefficients have been deleted.

Specifically, as follows:

“...spectra-texture feature scheme > texture feature scheme > spectra feature scheme; The accuracy ranking of the four growth stages is as follows: overwintering  > bolting  > flowering> podding...”

 “...In the spectra feature scheme, the six algorithms are ranked as follows: Xgboost>GBRT>Adaboost>RF>SVR>GPR; In the texture feature scheme, the six algorithms are ranked as follows: Adaboost>RF>GBRT>Xgboost>SVR>GPR; In the spectral-texture feature scheme, the six algorithms are ranked as follows: GBRT >RF >Xgboost > Adaboost > GPR >SVR ...

3) Ln 102, I am concerned that the five plants should be randomly selected. How the representativeness was determined?

Answer:

Thanks your suggestion !

The point sampling method is the five point sampling method. Randomly collect one plant from the middle and four diagonal positions of each plot.

4) Ln 124, In addition to the cited reference, describe the method of extracting image texture features here.

Answer:

Thanks your suggestion !

This detail has been added to the corresponding section.

Specifically, as follows:

“...Specifically, first, use the Python software OpenCV library to read the grayscale images of 138 bands; Secondly, use the numpy library to calculate 21 vegetation indices and generate corresponding grayscale images; Finally, use the numpy library and GLCM texture parameter calculation formula to obtain the texture parameters of each grayscale image...”

  • The discrepancy observed at high AGB values may be influenced by the sampling technique employed in this study, as only five plants were selected from each plot. I suggest including a more detailed explanation of this deviation in the discussion section to better clarify the model's performance in predicting higher AGB values. Additionally, Fig. 5 shows no clear trend of underestimation or overestimation, which implies that random sources of error may play a more significant role, please clarify.

Answer:

Thanks your suggestion !

The number of samples collected is consistent with many previous studies, which to some extent avoids the impact of measurement errors. Although three schemes have differences in predicting high AGB, the samples used by the three schemes are from same dataset. This means that the potential impact of random sampling on the three schemes is equal. 1: 1-line can clearly express the underestimation or overestimation of sample points, that is, the area above the 1:1 line is overestimated, and the area below the 1:1 line is underestimated. This study mainly emphasizes the applicability of narrowband spectral-texture features in estimating AGB, rather than the performance of the model itself. The study used six algorithms all aimed at evaluating the applicability of narrowband spectroscopy. In the discussion section, we focus more on comparing the performance of narrowband and broadband spectra, which is where our research differs from previous studies. The errors caused by the model itself have been well explained in previous research.

Reviewer 2 Report

Comments and Suggestions for Authors

"Estimating aboveground biomass of winter canola from UAV hyperspectral imagery using narrow-band spectral-texture features and machine learning" by Du et al. is a well-written manuscript. However, I have a few comments:

Introduction: The authors should present a comprehensive understanding of the surrounding literature. 

 Also, the literature review surrounding your topic is not well outlined.

Specify what it is and why it is important winter canola.

Material and methods: A map of the study area is helpful.

Results: Figures 2-5 are difficult to visualize.

Discussion: 

290-295: are previous studies only and would be welcome in the introduction 

302: What is the DJI Phantom 4? 

Conclusion: Write this section more succinctly.

Author Response

We thank the editors and reviewers for their hard work on this manuscript (plants-3166826). Your suggestions help to improved the quality of the manuscript. The parts of the manuscript that were revised have been marked in red. For the modified parts, the authors provide the following detailed explanation:

  • The authors should present a comprehensive understanding of the surrounding literature.Also, the literature review surrounding your topic is not well outlined. Specify what it is and why it is important winter canola

Answer:

Thanks your suggestion !

The layout of the introduction is as follows:

The first paragraph: Definition of aboveground biomass and the significance of remote sensing monitoring

Second paragraph: Research progress on remote sensing monitoring of aboveground biomass

Third paragraph: Limitations of remote sensing monitoring of aboveground biomass and advantages of unmanned aerial vehicles

The fourth paragraph: Introduce the objectives and research content of this study

In the study of estimating aboveground biomass using narrowband spectral texture features, winter rapeseed is only used as the object of evaluation, which does not mean that our method is only applicable to winter rapeseed. Therefore, in the introduction, we did not elaborate on winter rapeseed in detail, mainly focusing on remote sensing monitoring methods.

  • Material and methods: A map of the study area is helpful.

Answer:

Thanks your suggestion !

As you said, it would be great if a map of the research area could be provided. However, this hyperspectral drone is not equipped with a digital camera and cannot obtain a bird's-eye view of the entire experimental field from the air. We sincerely apologize for this suggestion.

  • Results: Figures 2-5 are difficult to visualize.

Answer:

Thanks your suggestion !

Figure 2 is the correlation analysis between spectral-texture information and winter canola AGB. Figure (a) shows the correlation coefficients between the spectral reflectance and corresponding texture parameters of 138 narrowband bands and aboveground biomass; Figure (b) shows the correlation coefficients between 21 vegetation indices and their corresponding texture parameters and aboveground biomass; The size of the correlation coefficient is mapped by color (blue to red indicates from low to high)

Figure 3 is the autocorrelation analysis of spectral and texture features. Figure (a) shows the autocorrelation analysis between the selected vegetation indices; Figure (b) shows the autocorrelation analysis between the selected texture parameters. The size of the correlation coefficient is mapped by color (blue to red indicates from low to high)

Figure 4 is the evaluation of the accuracy of winter canola AGB estimation. The three rows represent the accuracy results of estimating aboveground biomass for Scheme I, Scheme II, and Scheme III, respectively. The three columns represent the indicators for evaluating accuracy, namely R2, RMSE, and NRMSE. For each rotated bar chart, each set of bars represents a machine learning algorithm, and each bar represents a reproductive period.

④Figure 5 is the scatter plot of estimated and measured winter canola AGB values. The horizontal axis represents the measured aboveground biomass, and the vertical axis represents the estimated aboveground biomass. There are three shapes of sample points, corresponding to the results based on Scheme I, Scheme II, and Scheme III datasets. The color indicates the degree to which the sample points deviate from the 1:1 line.

  • Discussion: 290-295: are previous studies only and would be welcome in the introduction

Answer:

Thanks your suggestion !

The reason for citing this literature is that the sensor used in the literature is a broadband multispectral unmanned aerial vehicle, namely DJI Phantom 4. The research objective of this article is not remote sensing estimation of crop aboveground biomass. It may not be appropriate to include this article in the introduction.

  • 302: What is the DJI Phantom 4?

Answer:

Thanks your suggestion !

The DJI Phantom 4 is a unmanned drone equipped with 5 broadband multispectral lenses.

  • Conclusion: Write this section more succinctly

Answer:

Thanks your suggestion !

The content has been simplified

“...(2) After variable selection through autocorrelation analysis, 8 spectra feature variables and 23 texture feature variables were identified as the optimal spectral-texture feature combination, with correlation coefficients ranging from 0.63 to 0.72;

(3) Machine learning methods can establish robust models for estimating the winter canola AGB. The accuracy ranking of the six algorithms is as follows: Adaboost > RF> GBRT > Xgboost > GPR > SVR;

(4) The Adaboost model using the spectral-texture feature scheme performed the best in estimating the canola AGB. The combined use of narrowband spectra and image texture significantly improves the estimation accuracy . Compared to the spectra feature scheme, the model's R2 increased by 11.2%, RMSE decreased by 29%, and NRMSE decreased by 17%...”

Reviewer 3 Report

Comments and Suggestions for Authors

This is a well-structured study that makes an important contribution to the field of agricultural UAV remote sensing. Improvements to processing recommendations can enhance clarity, accessibility, and impact. The overall language is relatively smooth, but it is recommended to carefully proofread to eliminate small grammatical errors and spelling mistakes. I include some comments and suggestions to improve the article: 

Abstract:Lines 17-25,The introduction of the method is a bit verbose and needs to be further simplified.

Introduction

Line 47, The first appearance of the UAV requires the full name.UAV to Unmanned Aerial Vehicle(UAV)

 Material and methods

Lines 96-102, Could you add a picture to illustrate the location of the Experiment site?

Lines 137-145, How much sample size was used to train the machine learning model?

 Results

3.1 Which correlation analysis was used?This method should be described in the Methods section

3.2 Line 172-174, The rationale behind choosing a correlation coefficient threshold of 0.6 should be elaborated.  Explain why this specific threshold was chosen and how it compares to thresholds used in similar studies. The reduction from 159 to 18 spectral and 954 to 44 texture variables is substantial.  Could there be important information lost in this process?

3.3 Lines 201-221, The text mentions the accuracy rankings but does not provide any statistical analysis or metrics to support these rankings.  Consider including specific values (e.g., accuracy percentages) to substantiate the claims.  Furthermore, indicate whether statistical tests were conducted to determine the significance of differences between the schemes.

 Discussion

The discussion should delve deeper into the implications of your findings.  What does this mean for agricultural practices?  How could practitioners leverage these insights for better crop management? Additionally, consider discussing potential limitations of your study, such as the challenges of operationalizing hyperspectral imaging in field conditions or the need for extensive ground-truthing.

4.2 In Figure 7, while you indicate that broadband bands performed less effectively than narrowbands, it would be helpful to provide quantitative measures (e.g., R² values, RMSE) to support your claims.  Consider adding statistical analyses that compare the performance of different indices or bands to strengthen your conclusions.

 Conclusion

The ranking of algorithms based on their accuracy is useful. However, providing more details on the evaluation metrics used (e.g., cross-validation procedures, training/testing splits) would enhance the reliability of these results.

Comments on the Quality of English Language

The manuscript generally presents a logical structure, but certain sections could benefit from clearer transitions between ideas. Consider adding transitional phrases to enhance flow and coherence.Some sentences are overly complex and could be simplified. For instance, ensure that the main ideas are conveyed clearly without excessive use of jargon.

Author Response

We thank the editors and reviewers for their hard work on this manuscript (plants-3166826). Your suggestions help to improve the quality of the manuscript. The revised parts of the manuscript have been marked in red. For the modified parts, the authors provide the following detailed explanation:

  1. Abstract:Lines 17-25,The introduction of the method is a bit verbose and needs to be further simplified.

Answer:

Thanks your suggestion !

According to your suggestion, the simplified content is as follows:

Specifically, narrowband spectra and vegetation indices were extracted from the hyperspectral images. The Gray Level Co-occurrence Matrix (GLCM) method was employed to compute texture indices. Correlation analysis and autocorrelation analysis were utilized to determine the final spectral feature scheme, texture feature scheme, and spectral-texture feature scheme. Subsequently, machine learning algorithms  were applied to develop estimation models for winter canola biomass... ”

  1. Line 47, The first appearance of the UAV requires the full name.UAV to Unmanned Aerial Vehicle(UAV)

Answer:

Thanks your suggestion !

According to your suggestion, the revised content is as follows:

Satellites, Unmanned Aerial Vehicle (UAV), and near-ground remote sensing platforms can be utilized for the rapid, non-destructive acquisition of crop physiological and biochemical parameters, achieving the monitoring of crop growth indicators such as AGB

 Material and methods...

  1. Lines 96-102, Could you add a picture to illustrate the location of theExperiment site?

Answer:

Thanks your suggestion !

Reference 23 has provided the picture relevant to the location of the Experiment site. Duplicate images are not necessary to continue providing to avoid negative impact on academic ethics.

  1. Lines 137-145, How much sample size was used to train the machine learning model?

Answer:

Thanks your suggestion !

According to your suggestion, the number of collected has been added into the corresponding contents:

“...A total of 180 samples were collected...”

  1. Which correlation analysis was used? This method should be described in the Methods section?

Answer:

Thanks your suggestion !

Pearson correlation coefficient was used to measure the degree of linear correlation between two variables. The feature scheme was determined by sensitive analysis (Pearson correlation coefficient, r) between spectra/texture features and canola AGB. This method has been added into the Methods.

  1. 2 Line 172-174, the rationale behind choosing a correlation coefficient threshold of 0.6should be elaborated. Explain why this specific threshold was chosen and how it compares to thresholds used in similar studies ?The reduction from 159 to 18 spectral and 954 to 44 texture variables is substantial.  Could there be important information lost in this process?

Answer:

Thanks your suggestion !

In fact, this is a strategy that combines mathematical statistics and empirical analysis. In this sensitivity analysis, the significance level corresponding to a correlation coefficient greater than 0.4 is p<0.05, which means that 0.4 can be used as a threshold to screen variables. However, the threshold selects too many feature variables and the wavelength range of the used bands is relatively concentrated, which poses difficulties for subsequent autocorrelation analysis. We adjusted the threshold to 0.6 to increase the diversity of the wavelength range corresponding to the selected feature variables. For optical remote sensing to extract crop phenotype information, it often only requires several key wavelengths that can identify crop phenotype information, and not more is better.

  1. 3 Lines 201-221, The text mentions the accuracy rankings but does not provide any statistical analysis or metrics to support these rankings. Consider including specific values (e.g., accuracy percentages) to substantiate the claims. Furthermore, indicate whether statistical tests were conducted to determine the significance of differences between the schemes.

Answer:

Thanks your suggestion !

We have added statistical indicators to the ranking to objectively and straightforwardly demonstrate their rationality.The differences between the estimated results of different schemes have been visually demonstrated. The significance level of this difference reaches P<0.05.

The revised content is as follow:

“...Xgboost(R2=0.41~0.84;RMSE=77.54~3816.26kg/ha;NRMSE=17.42~26.01%)>GBRT(R2=0.41~0.84;RMSE=82.69~4040.94kg/ha;NRMSE=17.22~27.59%)>Adaboost(R2=0.46~0.82;RMSE=94.82~3998.51kg/ha;NRMSE=22.60~27.28%)>RF(R2=0.38~0.84;RMSE=86.93~3946.03kg/ha;NRMSE=22.79~26.92%)>SVR(R2=0.39~0.78;RMSE=93.55~4012.57kg/ha;NRMSE=19.76~27.38%)>GPR(R2=0.39~0.83;RMSE=112.44~4199.59kg/ha;NRMSE=23.61~30.32%); In the texture feature scheme, the ranking is as follows: Adaboost (R2=0.51~0.89; RMSE=73.37~3245.77kg/ha;NRMSE=18.40~22.02%)>RF(R2=0.51~0.85;RMSE=86.45~3120.87kg/ha;NRMSE=19.79~22.97%)>GBRT(R2=0.46~0.84;RMSE=88.05~3416.86kg/ha;NRMSE=23.08~25.58%)>Xgboost(R2=0.48~0.81;RMSE=86.00~3619.82kg/ha;NRMSE=23.10~26.49%)>SVR(R2=0.54~0.71;RMSE=91.59~3012.84kg/ha;NRMSE=20.38~29.06%)>GPR(R2=0.41~0.70;RMSE=86.48~4645.82kg/ha;NRMSE=22.67~31.82%); In the spectral-texture feature scheme, the ranking is as follows: GBRT (R2=0.49~0.86;RMSE=79.51~3005.50kg/ha;NRMSE=19.63~26.17%)>RF(R2=0.51~0.89;RMSE=84.19~2924.98kg/ha;NRMSE=19.78~22.08%)>Xgboost(R2=0.48~0.85;RMSE=88.41~3044.32kg/ha;NRMSE=19.28~23.65%)>Adaboost(R2=0.41~0.84;RMSE=77.54~3816.26kg/ha;NRMSE=17.42~26.01%)>GPR(R2=0.43~0.77;RMSE=92.14~3645.81kg/ha;NRMSE=23.41~27.02%)> SVR(R2= 0.51~0.62; RMSE= 91.57~3239.99 kg/ha;NRMSE= 18.83~29.06 %)...”

  1. The discussion should delve deeper into the implications of your findings. What does this mean for agricultural practices?  How could practitioners leverage these insights for better crop management? Additionally, consider discussing potential limitations of your study, such as the challenges of operationalizing hyperspectral imaging in field conditions or the need for extensive ground-truthing.

Answer:

Thanks your suggestion !

We have added section 4.4 to the discussion section to elaborate on the issues you have raised:

“...In this study, we evaluated the potential application of narrowband spectra-texture information provided by hyperspectral images in estimating the winter canola aboveground biomass. Compared to the broadband spectra-texture information provided by widely used multispectral images, hyperspectral images have clear advantages in improving the accuracy of aboveground biomass estimation, providing an important data source for extracting temporal characteristics of aboveground biomass during winter canola growth, and can be further applied to winter canola yield prediction and field irrigation-nitrogen supply management evaluation. However, the estimation models established by machine learning driven spectra-texture information also have certain potential application limitations. On the one hand, data-driven methods have weak theoretical rationality, and the established estimation models lack universality and are only applicable to specific samples. This results in the selected optimal spectra-texture feature combination and modeling algorithm not necessarily being applicable to all crops. When the observation conditions (e.g.sun-sensor-target geometric optical conditions, canopy structure, and soil background) change, it is necessary to re-determine the appropriate features combination and modeling methods. On the other hand, the robustness of the estimation model requires a certain number of samples. For low altitude remote sensing, collecting sufficiently representative samples on small-scale farmland is more difficult than that at regional scales. In the future, it is necessary to further explore the applicability of transfer learning strategies in extracting crop aboveground biomass information from low altitude remote sensing to reduce the modeling samples demand. Specifically, it is possible to attempt pre training estimation models using existing publicly available large-scale farmland datasets or theoretical spectra datasets, and then transferring them to estimate farmland crop aboveground biomass...”

  1. 2 In Figure 7, while you indicate that broadband bands performed less effectively than narrowbands, it would be helpful to provide quantitative measures (e.g., R² values, RMSE) to support your claims. Consider adding statistical analyses that compare the performance of different indices or bands to strengthen your conclusions.

Answer:

Thanks your suggestion !

Accroding to your suggestion,the quantitative indices have been added into the corresponding part: 

“...Figure 7 shows that the ability of broadband to capture changes in winter canola AGB is lower than that of narrowband. For example, the correlation coefficients for the broadband and narrowband spectra texture of the blue band are concentrated in the ranges of 0.49-0.63 and 0.50~0.69, respectively (Figure 7a). Narrowband spectral information can obtain the most sensitive areas of changes in winter rapeseed biomass, while broadband spectral information will blur the relationship between the two. For example, for the Red-edge band, the correlation coefficient between broadband spectra and canola AGB is 0.55. Compared to broadband spectra, the correlation coefficient between narrowband spectra with a wavelength of 726 nm and Canola AGB is 0.65 (Figure 7d)...”

  1. The ranking of algorithms based on their accuracy is useful. However, providing more details on the evaluation metrics used (e.g., cross-validation procedures, training/testing splits) would enhance the reliability of these results.

Answer:

Thanks your suggestion !

Accroding to your suggestion, the quantitative indices have been added into the conclusion. The modeling details have been added into the Method:

“...(3) Machine learning methods can establish robust models for estimating the winter canola AGB. The accuracy ranking of the six algorithms is as follows: Adaboost (R2=0.82~0.91;RMSE=1710.79~2408.43 kg/ha;NRMSE=19.88~24.07 %) > RF(R2= 0.84~0.89; RMSE= 1993.29~2392.10 kg/ha;NRMSE= 21.58~23.97 %)> GBRT (R2= 0.84~0.87; RMSE= 2274.08~2296.47 kg/ha;NRMSE= 23.26~23.40 %)> Xgboost (R2= 0.81~0.85;RMSE= 2246.95~2328.48 kg/ha;NRMSE= 23.10~23.64%) > GPR (R2 = 0.64~0.83; RMSE= 2351.82~2900.25 kg/ha;NRMSE= 23.73~27.02%) > SVM (R2 = 0.62~0.78; RMSE= 2622.88~3239.99 kg/ha;NRMSE= 25.35~29.06 %);..”

“...K-fold cross validation is used to avoid overfitting problems encountered during model training.The ratio of the number of samples in the modeling set to the validation set was 7:3...”

  1. The manuscript generally presents a logical structure, but certain sections could benefit from clearer transitions between ideas. Consider adding transitional phrases to enhance flow and coherence.Some sentences are overly complex and could be simplified. For instance, ensure that the main ideas are conveyed clearly without excessive use of jargon.

Answer:

Thanks your suggestion !

Based on your suggestion, we have reviewed the entire text and made the following modifications:

...Compared to traditional measurement, optical remote sensing technology can obtain information on the crop physiological and biochemical properties by capturing crop canopy spectra, thus becomes a popular and promising monitoring method...

 ...Satellites, Unmanned Aerial Vehicle (UAV), and near-ground remote sensing platforms can be utilized for the rapid, non-destructive acquisition of crop growth indicators such as AGB[5]... 

...However, the capability of vegetation index on the crop AGB estimation is limited due to the saturation effect caused by dense canopy [17]. This phenomenon can be alleviated by using narrowband vegetation indices from hyperspectral images [18]. Besides spectra information, ultra-high ground resolution UAV images (1cm or higher) provides abundant image texture information for crop phenotyping studies [19]...

...Liu et al. combined multispectral vegetation indices with texture parameter to achieve reasonable winter canola nitrogen nutrition index (NNI) estimation [21]...

...The corresponding texture parameters can be used to describe a specific object more objectively and reliably [22]...

Reviewer 4 Report

Comments and Suggestions for Authors

The work is interesting from the applied point of view. Its main advantage is the results of analyzing a large amount of information. This also determines the main disadvantage, which is the use of standard machine learning methods. In this regard, it should be noted that the novelty of the work is determined by the object of research. 

I believe that it would be advisable to remove “UAV” from the title. This will shorten the title and draw the reader's attention to the main thing - hyperspectral images, the processing of which the study is devoted to. 

Please pay attention to Figure 1 and correct it.

Author Response

  1. I believe that it would be advisable to remove “UAV” from the title. This will shorten the title and draw the reader's attention to the main thing - hyperspectral images, the processing of which the study is devoted to.

Answer:

Thanks your suggestion !

Accroding to your suggestion,the revised title is as follow:

“Estimating winter canola aboveground biomass from  hyperspectral images using narrowband spectral-texture features and machine learning”

  1. Please pay attention to Figure 1 and correct it.

Answer:

Thanks your suggestion !

We have modified some of the textual descriptions in Figure 1 to avoid readers' misunderstanding of the research process:

Figure 1 Technology roadmap.

Round 2

Reviewer 2 Report

Comments and Suggestions for Authors

I ask the authors to read my previous comments carefully and change the text accordingly. Their answers are not satisfactory at all.

Author Response

We thank the editors and reviewers for their hard work on this manuscript (plants-3166826). Your suggestions help to improve the quality of the manuscript. The revised parts of the manuscript have been marked in red. For the modified parts, the authors provide the following detailed explanation:

1) The authors should present a comprehensive understanding of the surrounding literature. Also, the literature review surrounding your topic is not well outlined. Specify what it is and why it is important winter canola.

Answer:

Thanks your suggestion !

We added information about the importance and research necessity of winter rapeseed in the last paragraph of the introduction to correspond with the title of the manuscript:

“...Canola is one of the important oil crops in China. Under the background of global climate change, the planting range of canola is showing a trend of moving northward and widely planted in the northwest region. For this,we aim to investigate the potential of ultra-high spatial resolution UAV hyperspectral images in estimating winter canola biomass ...”

The layout of the introduction is as follows:

The first paragraph: Definition of aboveground biomass and the significance of remote sensing monitoring

Second paragraph: Research progress on remote sensing monitoring of aboveground biomass

Third paragraph: Limitations of remote sensing monitoring of aboveground biomass and advantages of unmanned aerial vehicles

The fourth paragraph: Introduce the objectives and research content of this study

2) Material and methods: A map of the study area is helpful.

Answer:

Thanks your suggestion !

As you said, it would be great if a map of the research area could be provided. However, this hyperspectral drone is not equipped with a digital camera and cannot obtain a bird's-eye view of the entire experimental field from the air. We sincerely apologize for this suggestion.Meanwhile,Reference 23 has provided the picture relevant to the location of the Experiment site. Duplicate images are not necessary to continue providing to avoid negative impact on academic ethics.

3) Results: Figures 2-5 are difficult to visualize.

Answer:

Thanks your suggestion !

Change subheadings and vertical and horizontal axis names for easier reading:

Figure 3 Autocorrelation analysis of spectral and texture features

Figure 5 Scatter plot of estimated and measured winter canola AGB values

The meanings conveyed by these pictures are as follows:

Figure 2 is the correlation analysis between spectral-texture information and winter canola AGB. Figure (a) shows the correlation coefficients between the spectral reflectance and corresponding texture parameters of 138 narrowband bands and aboveground biomass; Figure (b) shows the correlation coefficients between 21 vegetation indices and their corresponding texture parameters and aboveground biomass; The size of the correlation coefficient is mapped by color (blue to red indicates from low to high);

Figure 3 is the autocorrelation analysis of spectral and texture features. Figure (a) shows the autocorrelation analysis between the selected vegetation indices; Figure (b) shows the autocorrelation analysis between the selected texture parameters. The size of the correlation coefficient is mapped by color (blue to red indicates from low to high);

Figure 4 is the evaluation of the accuracy of winter canola AGB estimation. The three rows represent the accuracy results of estimating aboveground biomass for Scheme I, Scheme II, and Scheme III, respectively. The three columns represent the indicators for evaluating accuracy, namely R2, RMSE, and NRMSE. For each rotated bar chart, each set of bars represents a machine learning algorithm, and each bar represents a reproductive period;

Figure 5 is the scatter plot of estimated and measured winter canola AGB values. The horizontal axis represents the measured aboveground biomass, and the vertical axis represents the estimated aboveground biomass. There are three shapes of sample points, corresponding to the results based on Scheme I, Scheme II, and Scheme III datasets. The color indicates the degree to which the sample points deviate from the 1:1 line.
